# Infection with endosymbiotic *Spiroplasma* disrupts tsetse (*Glossina fuscipes fuscipes*) metabolic and reproductive homeostasis

**Jae Hak Son**[1☯], **Brian L. Weiss**[1☯]*, **Daniela I. Schneider**[1¤a], **Kiswend-sida M. Dera**[2,3],
**Fabian Gstöttenmayer**[2], **Robert Opiro**[4], **Richard Echodu**[4], **Norah P. Saarman**[5¤b],
**Geoffrey M. Attardo**[1¤c], **Maria Onyango**[1¤d], **Adly M. M. Abd-Alla**[2], **Serap Aksoy**[1]*

**1** Department of Epidemiology of Microbial Diseases, Yale School of Public Health, New Haven, Connecticut, United States of America, **2** Insect Pest Control Laboratory, Joint FAO/IAEA Programme of Nuclear Techniques in Food and Agriculture, Vienna, Austria, **3** Insectarium de Bobo-Dioulasso—Campagne d'Eradication de la mouche Tse´-tse´ et de la Trypanosomiase (IBD-CETT), Bobo-Dioulasso, Burkina Faso, **4** Department of Biology, Faculty of Science, Gulu University, Gulu, Uganda, **5** Department of Ecology and Evolutionary Biology, Yale University, New Haven, Connecticut, United States of America

☯ These authors contributed equally to this work.
¤a Current address: Moffit Cancer Center, Tampa, Florida, United States of America
¤b Current address: Department of Biology, Utah State University, Logan, Utah, United States of America
¤c Current address: Department of Entomology and Nematology, University of California, Davis, United States of America
¤d Current address: Wadsworth Centre, New York State Department of Health, Slingerlands, New York, United States of America
* brian.weiss@yale.edu (BLW); serap.aksoy@yale.edu (SA)

**Data Availability Statement:** Most of the data are contained within the manuscript and/or supporting information files. RNA-seq data has been submitted to NCBI, GEO accession number

## Abstract

Tsetse flies (*Glossina* spp.) house a population-dependent assortment of microorganisms that can include pathogenic African trypanosomes and maternally transmitted endosymbiotic bacteria, the latter of which mediate numerous aspects of their host's metabolic, reproductive, and immune physiologies. One of these endosymbionts, *Spiroplasma*, was recently discovered to reside within multiple tissues of field captured and laboratory colonized tsetse flies grouped in the Palpalis subgenera. In various arthropods, *Spiroplasma* induces reproductive abnormalities and pathogen protective phenotypes. In tsetse, *Spiroplasma* infections also induce a protective phenotype by enhancing the fly's resistance to infection with trypanosomes. However, the potential impact of *Spiroplasma* on tsetse's viviparous reproductive physiology remains unknown. Herein we employed high-throughput RNA sequencing and laboratory-based functional assays to better characterize the association between *Spiroplasma* and the metabolic and reproductive physiologies of *G. fuscipes fuscipes* (*Gff*), a prominent vector of human disease. Using field-captured *Gff*, we discovered that *Spiroplasma* infection induces changes of sex-biased gene expression in reproductive tissues that may be critical for tsetse's reproductive fitness. Using a *Gff* lab line composed of individuals heterogeneously infected with *Spiroplasma*, we observed that the bacterium and tsetse host compete for finite nutrients, which negatively impact female fecundity by increasing the length of intrauterine larval development. Additionally, we found that when males are infected with *Spiroplasma*, the motility of their sperm is compromised following transfer to the female spermatheca. As such, *Spiroplasma* infections appear to adversely impact male

GSE183197 and BioProject accession number PRJNA759598.

**Funding:** Funding was generously provided by the NIH/NIAID (RO1AI068932), the Ambrose Monell Foundation (monellfoundation.org), and the Li Foundation (lifoundation.org) to SA. The funders had no role in study design, data collection and analysis, decision to publish, or preparation of the manuscript.

**Competing interests:** The authors have declared that no competing interests exist.

reproductive fitness by decreasing the competitiveness of their sperm. Finally, we determined that the bacterium is maternally transmitted to intrauterine larva at a high frequency, while paternal transmission was also noted in a small number of matings. Taken together, our findings indicate that *Spiroplasma* exerts a negative impact on tsetse fecundity, an outcome that could be exploited for reducing tsetse population size and thus disease transmission.

## Author summary

Endosymbiotic bacteria regulate numerous aspects of their host's reproductive physiology. Natural populations of the tsetse fly, *Glossina fuscipes fuscipes* (*Gff*), house heterogeneous infections with the bacterium *Spiroplasma glossinidia*. Infection with the bacterium results in the presentation of several phenotypes in both male and female *Gff* that would put them at a significant reproductive disadvantage when compared to their counterparts that do not house the bacterium. These *Spiroplasma* induced phenotypes include changes in sex–biased gene expression in the reproductive organs, a depletion in the availability of metabolically critical lipids in pregnant females that results in delayed larval development, and compromised sperm fitness. These findings indicate that *Spiroplasma* exerts an overall negative impact on both male and female reproductive fitness and thus likely has a profound effect on fly population structure. This outcome, in conjunction with the fact that *Spiroplasma* infected tsetse are unusually refractory to infection with pathogenic African trypanosomes, indicates that the bacterium could be experimentally exploited to reduce disease transmission through the fly.

## Introduction

Tsetse flies (*Glossina spp.*), which vector pathogenic African trypanosomes, reproduce via a process called adenotrophic viviparity. Following mating, female tsetse ovulate one oocyte per gonotrophic cycle (GC). The oocyte is fertilized in the maternal uterus by sperm that are released from the spermathecae. Unlike in most arthropods, tsetse embryos and larvae develop exclusively *in utero*, and larvae receive nourishment in the form of maternal milk secretions. Following the completion of larvigenesis, the mother gives birth to a fully developed 3rd instar larva that pupates within 30 minutes. This process repeats itself approximately every 10 days [1]. During the mating process, male tsetse transfer seminal fluid (SF), which contains sperm as well as numerous male accessory gland (MAG) derived proteins, into the reproductive tract of receptive females. Once in the female's uterus, this mixture forms into a proteinaceous spermatophore that facilitates successful transfer of sperm into the spermathecae for long-term storage [2–6]. Because tsetse's viviparous reproductive strategy results in the production of relatively few offspring, targeting reproduction can be a most effective approach to reduce tsetse populations. Methods to inhibit tsetse fecundity can be highly effective in reducing disease transmission given that the fly is an obligate vector for parasite transmission [1,7]. In fact, the application of sterile male programs has been very successful with tsetse [8] and is currently endorsed to eliminate tsetse populations on the African continent [8,9].

Tsetse flies have evolved long-term associations with several vertically transmitted endosymbiotic bacteria that impact numerous aspects of their host's nutritional, developmental and reproductive physiology. All tsetse flies house the obligate endosymbiont *Wigglesworthia*,

which provides nutrients absent from the fly's vertebrate blood-specific diet [10–13]. In addition to *Wigglesworthia*, laboratory reared and natural tsetse populations can also harbor facultative *Sodalis* as well as parasitic *Wolbachia* and *Spiroplasma* [14,15]. To date *Spiroplasma* infections in tsetse have been found exclusively within tsetse flies of the subgenus Palpalis [14,15]. In both field captured and laboratory reared *Glossina fuscipes fuscipes* (*Gff*), the bacterium resides in reproductive and digestive tissues as well as hemolymph [14]. *Spiroplasma* induces an immune protective effect in laboratory reared *Gff*, as flies that house the bacterium are significantly more resistant to infection with trypanosomes than are flies that do not [15]. The mechanism that underlies *Spiroplasma* enhanced refractoriness to trypanosome infection in *Gff* is currently unknown. However, in other insects the bacterium confers pathogen (e.g., nematodes, fungi and parasitoid wasps) resistant phenotypes through the production of immune effector molecules and/or through nutrient scavenging that limits metabolically critical nutrients for other pathogens [16–19].

Infection with *Spiroplasma* can also impact host reproductive fitness and lead to reproductive abnormalities. In *Drosophila*, females infected with *Spiroplasma* are less fecund and produce fewer eggs, which may be a consequence of nutritional competition between the fly and bacterium for metabolically important free (hemolymph-borne) lipids [20]. In several insect taxa, the bacterium induces a male killing phenotype [21–23]. The potential functional role of *Spiroplasma* in tsetse reproductive physiology is currently unknown. Herein we performed a detailed investigation of the molecular and physiological associations that characterize reproductive aspects of the *Gff-Spiroplasma* symbiosis. Specifically, we performed RNA sequencing (RNA-seq) of reproductive tissues from field captured (Uganda) *Spiroplasma* infected and uninfected male and female *Gff* and report on genes and pathways that are differentially regulated in the presence of the bacterium. We also made use of a *Gff* lab line that carries a heterogenous infection with *Spiroplasma* to characterize the trans-generational transmission dynamics of this endosymbiont and to characterize *Spiroplasma*-induced metabolic and reproductive phenotypes in tsetse. Knowledge obtained from this study provides insight into the physiological mechanisms that underlie the tsetse-*Spiroplasma* symbiosis and may have translational implications with respect to controlling tsetse populations and the ability of the fly to transmit African trypanosomes.

## Results

### Overall gene expression profile

To understand the potential effects of *Spiroplasma* infection on tsetse's reproductive physiology, we compared the RNA-seq data obtained from the male reproductive organs (testis, accessory gland and ejaculatory duct) of *Spiroplasma* infected (*Gff*^Spi+) and uninfected (*Gff*^Spi-) individuals obtained from wild populations in Uganda. We similarly obtained and compared RNA-seq data from *Gff*^Spi+ and *Gff*^Spi- female reproductive organs (ovaries and oocytes) isolated from the same populations. The percentage of reads from the different biological replicates that mapped to the *Gff* reference genome ranged from 49% to 81% (S1 Table in S1 Appendix).

We used principal component (PC) and hierarchical cluster (HC) analyses to compare the overall gene expression profiles across all biological replicates according to sex and *Spiroplasma* infection status (Fig 1A and 1B). We found that PC1 and PC2 accounted for 80% and 7% of variance across all biological replicates, respectively. The variance in PC1 could be explained by differences in sex, with females clustering on the left side along the PC1 axis, and males clustering on the right side along the PC1 axis (Fig 1A). HC analyses also demonstrated a similar clustering result among different biological replicates (Fig 1B). Female and male

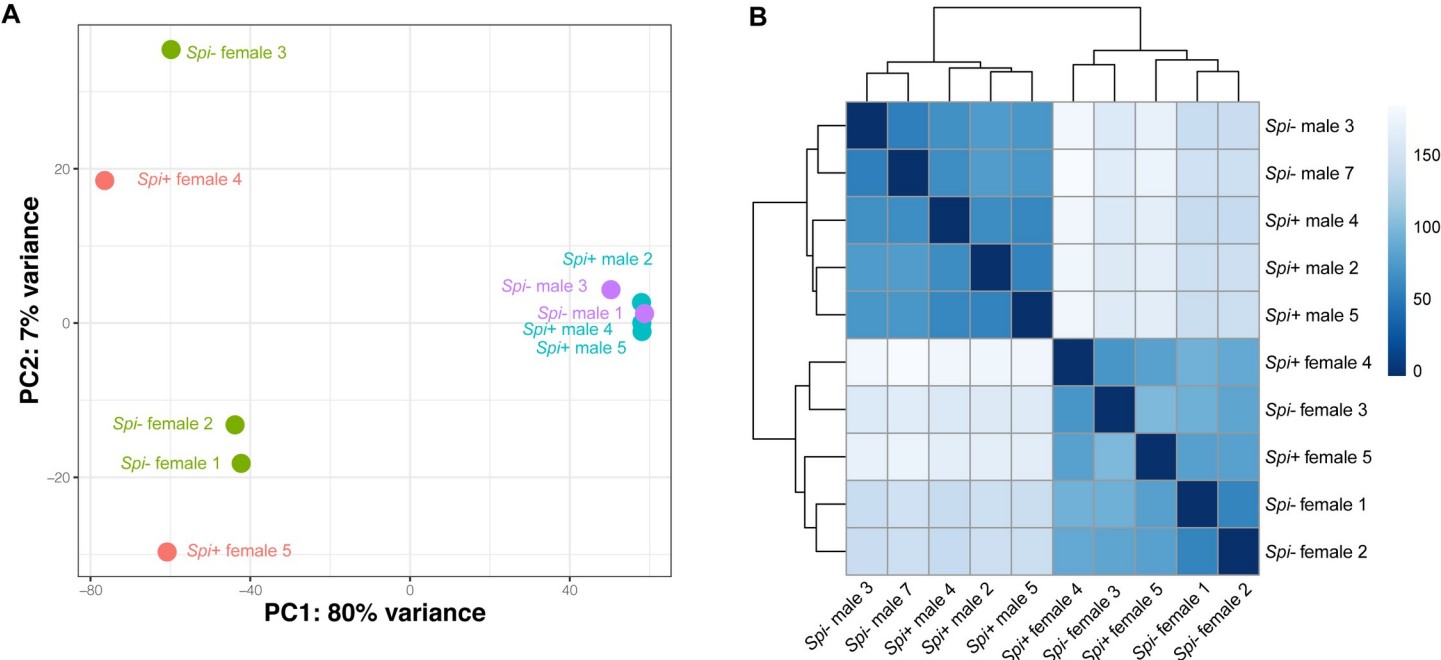

**Fig 1.** Principal component analysis (A) and Hierarchical clustering (B) of expression data. The Principal Component Analysis (PCA) is based on differentially expressed genes from the female and male samples, while the Hierarchical Clustering (HC) reflects all genes within the dataset. *Spiroplasma*-infection status and sex are color-coded in the PCA.

clusters were distant from each other, and within the male samples, $Gff^{Spi+}$ were separated from the $Gff^{Spi-}$ samples. However, within the female samples, one $Gff^{Spi+}$ (female 4) and one $Gff^{Spi-}$ (female 3), were distant to the other three female samples (one $Gff^{Spi+}$ and two $Gff^{Spi-}$). This difference we noted within the female dataset could be due to the presence of greater variation in gene expression within the female reproductive tissues when compared to males, which is similar to the results shown in our PCA.

## The impact of *Spiroplasma* on sex-biased gene expression in males and females

As our biological replicates clustered by sex, we first quantified the proportion of differentially expressed (DE) genes between females and males (sex-biased genes) using sex as a factor in the model we created in DESeq2 (see Materials and Methods for details). Of the 15,247 genes annotated in the *Gff* genome, we detected 10,540 genes expressed in our transcriptomes, with an adjusted $p$-value created by FDR correction (S1 Data) [24]. We observed that 21.7% of these genes (2,288/10,540) were preferentially expressed in females (female-biased genes; $\log_2$male/female<0 and adjusted $p<0.05$), while 20.5% (2,164/10,540) were preferentially expressed in males (male-biased genes; $\log_2$male/female>0 and adjusted $p<0.05$). Using pairwise comparisons for *Spiroplasma* infection status within each sex from the DESeq2 model, we found that a total of 194 and 299 genes were DE upon *Spiroplasma* infection in males and females, respectively. Within the male dataset, we determined that 117 (60.3%) and 77 (39.7%) of the DE genes were up- and down-regulated upon infection (adjusted $p<0.05$), respectively (Fig 2A, S2 Table in S1 Appendix and S2 Data). A similar analysis of the 299 DE genes in females showed that 61 (21.4%) and 238 (79.6%) were up- and down-regulated in the presence of *Spiroplasma*, respectively (Fig 2B and S3 Table in S1 Appendix).

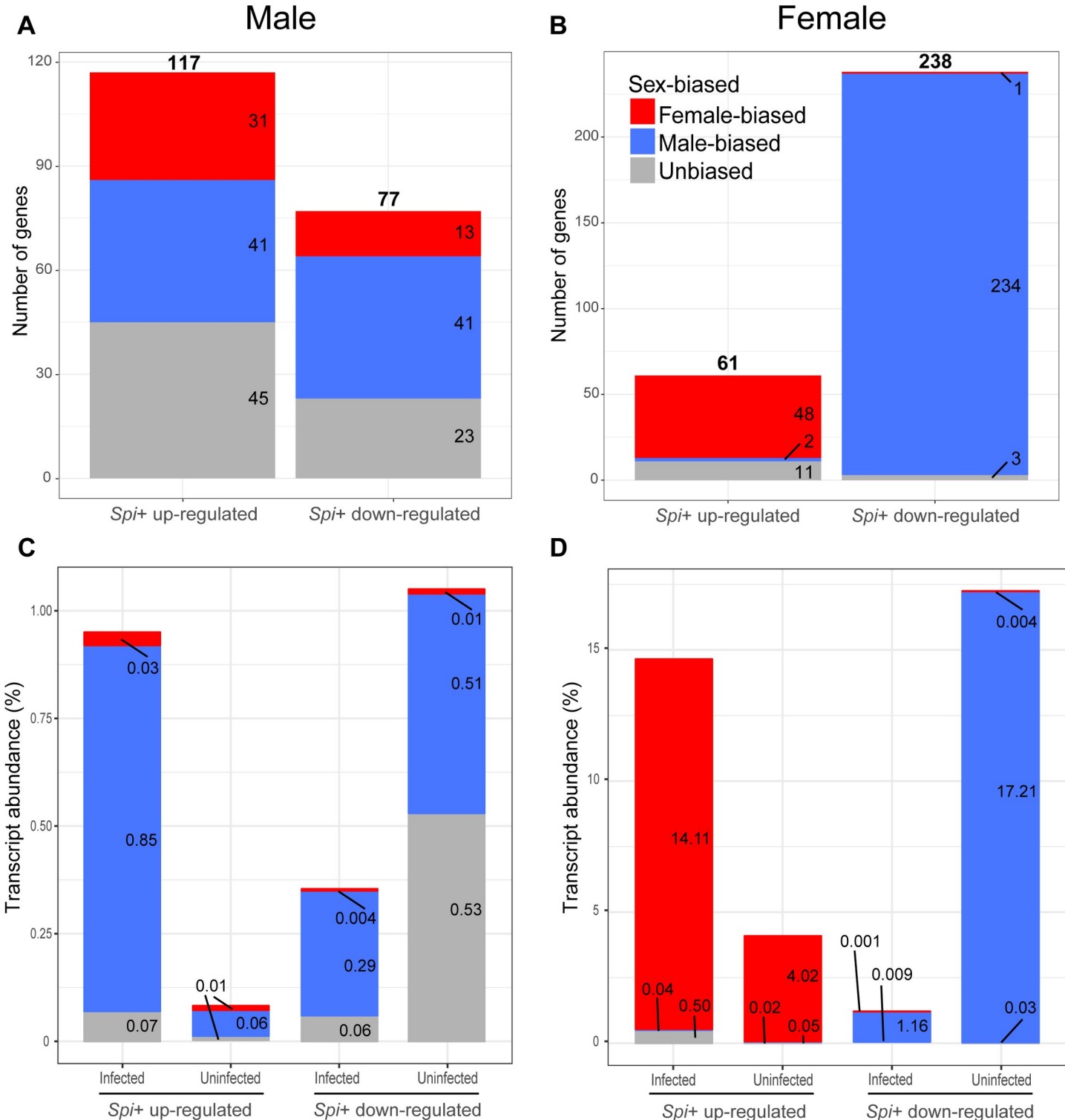

**Fig 2. Enrichment of genes with sex-biased expression that are differentially expressed between *Gff*^*Spi+*^ and *Gff*^*Spi-*^ flies.** (A) The number of genes with sex-biased expression that are up- and down-regulated in *Gff*^*Spi+*^ males. (B) The number of genes with sex-biased expression profiles that are up- and down-regulated in *Gff*^*Spi+*^ females. (C) Transcript abundance (%) of up- and down-regulated genes relative to total transcript abundance in *Gff*^*Spi+*^ and *Gff*^*Spi-*^ males. (D) Transcript abundance of up- and down-regulated genes relative to total transcript abundance in *Gff*^*Spi+*^ and *Gff*^*Spi-*^ females. Female-biased genes are represented as red blue and male-biased genes as blue. Genes with no sex-bias are indicated as gray.

*Spiroplasma* infections can manipulate the reproductive physiology of their host insects [25]. In *Drosophila*, infections with the *S. poulsonii* strain MSRO affects gene regulation and dosage compensation machinery in males, resulting in male-killing during embryogenesis [22,26]. Because genes displaying sex-biased expression profiles in reproductive tissues can confer sexual phenotypes [27,28], we first identified the sex-biased genes expressed in male and female gonads, and next evaluated their DE status based on *Spiroplasma* infection. We determined that 41 of the 117 genes (35%) up-regulated, and 41 of the 77 genes (53%) down-regulated in $Gff^{Spi+}$ males are male-biased (Fig 2A and S2 Table in S1 Appendix). We also noted that the DE male-biased genes ($n = 82$) identified within the male DE gene set ($n = 194$) are enriched when compared to genome-wide male-biased genes ($n = 2164$) (Fig 2A and S2 Table S1 Appendix; $p < 0.001$ for up-regulation and $p < 10^{-9}$ for down-regulation in Fisher's exact test). We next examined the transcript abundance of the 41 up-regulated male-biased genes and determined their contribution to be 0.85% of the total transcriptome in the infected group, up from the 0.06% in the uninfected group (Fig 2C). We similarly analyzed the 41 down-regulated male-biased genes and determined their contribution to be 0.29% of the total transcriptome in the infected group, down from the 0.51% in the uninfected group (Fig 2C).

We next performed the same analysis with the female DE gene dataset. We determined that 48 of the 61 genes (78%) up-regulated in the $Gff^{Spi+}$ females were female-biased and enriched within the up-regulated genes when compared to genome-wide female-biased genes ($n = 2288$) (S3 Table in S1 Appendix; $p < 10^{-20}$ in Fisher's exact test). Conversely, only one of the 238 genes down-regulated in $Gff^{Spi+}$ females was female-biased (Fig 2B and S3 Table in S1 Appendix), while 234 (98%) were male-biased (Fig 2B and S3 Table in S1 Appendix). The male-biased genes were thus overrepresented within the down-regulated genes (S3 Table in S1 Appendix; $p < 10^{-149}$ in Fisher's exact test). We also examined the transcript abundance of the 48 up-regulated female-biased genes and determined their contribution to be 14.11% of the total transcriptome in the infected group, up from the 4.02% transcript abundance in the uninfected group (Fig 2D). The transcript abundance of the 234 down-regulated male-biased genes in the female dataset represented 1.16% of the total transcriptome in the infected group, down from 17.21% transcript abundance in the uninfected group (Fig 2D).

In both sexes, unbiased (non-sex biased) genes were underrepresented in the DE datasets when compared to the genome-wide unbiased genes (Fig 2, S2 and S3 Tables in S1 Appendix; $p < 10^{-4}$ for males and $p < 10^{-9}$ for females in Fisher's exact test). Collectively, these results suggest that *Spiroplasma* infections strongly influence the expression of sex-biased genes in tsetse's reproductive tissues. Given that sexual dimorphisms are largely shaped by sex-biased gene expressions [27,28], the significant number of sex-biased genes and transcript abundance affected by *Spiroplasma* in females relative to males suggests that the endosymbiont may affect female reproductive physiology to a greater extent.

### *Spiroplasma* infection effects on genes that encode spermatophore constituents

When tsetse copulate, the ejaculate, which is composed of sperm and secretory products (i.e., seminal fluid) derived from the male testes and accessory glands, are transferred to the female uterus where they are encapsulated into a spermatophore. The spermatophore functions as a protective container for the ejaculate and ensures that sperm and seminal fluid are delivered to the female spermathecal ducts, which in turn modifies female behavior, including potential inhibition of further matings by competing males [3–5]. Within 24 h of the commencement of copulation, sperm are transferred to the female's spermatheca and the spermatophore is discharged from the uterus [3]. We had previously identified the proteins detected in the *G.*

*morsitans morsitans (Gmm)* spermatophore by collecting the structures from the female spermatheca shortly after completion of copulation [2]. We first identified the homologs of these *Gmm* protein encoding genes in *Gff* and then evaluated their expression status in the presence of *Spiroplasma*. We found that of the 287 genes whose products were detected in the *Gmm* spermatophore, seven were DE in *Gff*$^{Spi+}$ males and 12 were DE in *Gff*$^{Spi+}$ females relative to *Gff*$^{Spi-}$ datasets (S1 Data). Of the seven genes encoding spermatophore proteins contributed by male gonads, six were up-regulated and one was down-regulated in the presence of *Spiroplasma*. Of the 12 genes encoding spermatophore proteins contributed by female reproductive organs, four were up-regulated and eight were down-regulated in the presence of *Spiroplasma*. Among the DE genes in females were two that encode transcription factors and three that encode serine protease inhibitors, while the up-regulated genes in males primarily encoded cuticle related proteins.

## Immunity genes up-regulated in *Gff*$^{Spi+}$ males

Long term persistence of endosymbionts requires evasion of, and/or resistance against, host immune responses. Although the absence of metabolically costly immune responses would benefit host fitness, induced immunity can also be beneficial to the symbiosis as it confers resistance to other pathogens that could compete for limited nutritional resources. In *Drosophila*, *Spiroplasma* infections do not activate host immune responses, and the bacterium is not susceptible to either the cellular or humoral arms of the fly's immune system [29]. To understand tsetse-*Spiroplasma* immune dynamics, we investigated whether DE genes in *Gff*$^{Spi+}$ males and females encoded immunity related functions. We found that in the male gonads, the presence of *Spiroplasma* significantly induces Toll pathway constituents, including *toll-like receptor 7* (GFUI009072), the antimicrobial peptide *defensin* (GFUI031425), and *easter* (GFUI050711), which encodes a serine-protease required to process the extracellular Toll ligand Spätzle. We also noted that in addition to the Toll signaling pathway, several abundantly expressed Mucin encoding genes (GFUI039642, GFUI016405, GFUI054349, GFUI017943) were induced in *Gff*$^{Spi+}$ males. Mucins are produced by epithelial tissues where they function in different roles from lubrication to cell signaling to forming chemical barriers as well as binding and entrapping pathogens [30]. Taken together, this analysis indicates that *Spiroplasma* infection induces the upregulation of a highly restricted repertoire of immune related genes in the reproductive tract of *Gff* males. The bacterium had no impact on immune gene expression in *Gff* females.

## Gene ontology (GO) analysis of DE gene products

To understand the major putative function(s) for the 493 DE gene products associated with *Spiroplasma* infection in *Gff* reproductive physiology, we performed GO (gene ontology) term enrichment analysis [31]. We found that within the up-regulated genes in *Gff*$^{Spi+}$ males, the significantly enriched GO terms are associated with cuticle development, chitin process, cell adhesion, defense response and receptor activity (S1A Fig and S3 Data). The down-regulated genes in *Gff*$^{Spi+}$ males were enriched for lipid catabolic process and peptidyl-dipeptidase activity (S1B Fig and S3 Data). The up-regulated genes in *Gff*$^{Spi+}$ females were enriched for lipid metabolic process and defense response (S1C Fig and S3 Data) while the down-regulated genes in *Gff*$^{Spi+}$ females were enriched for peptidase activity, proteolysis, and chitin metabolic process (S1D Fig and S3 Data).

## *Spiroplasma* infection effects on female gene expression

The up-regulated female-biased genes in *Gff*$^{Spi+}$ females comprised over 14% of the entire transcriptome, while these same genes comprised only 4% of the total transcriptome in *Gff*$^{Spi-}$

females. Among the up-regulated gene dataset, we noted the presence of several highly abundant tsetse milk proteins, including Mgp1 (GFUI006902), Mgp10 (GFUI050429), Mgp4 (GFUO050451). In addition, we detected high level expression of a gene annotated as Apolipoprotein (GFUI006901) located adjacent to the *mgp1* (GFUI006902) locus.

We also noted that 234 male-biased genes were dramatically down-regulated in *Spiroplasma* infected females. These genes contributed a total of 1% of the transcriptome in the infected state, down from 17% in the uninfected state. Among the abundant and highly reduced genes were ones that encode multiple proteins annotated with digestive functions, such as those that had signatures of midgut trypsins (GFUI029963, GFUI029966, GFUI14886, GFUI024126, GFUI049688, GFUI006483, GFUI026212), serine proteases (GFUI026465, GFUI009869, GFUO026202), trypsin-like serine proteases (GFUI028730, GFUI028738), zinc-carboxypeptidases (GFUI030548, GFUI007477), chymotrypsin-like proteins (GFYI032994, GFUI032998, GFUI010855), and a carboxypeptidase (GFUI022995). The physiological consequences of this reduced proteolytic activity on $Gff^{Spi+}$ females remain to be determined.

## Impact of *Spiroplasma* infection on *Gff* female metabolism and reproductive fitness

We next investigated whether *Spiroplasma* induced differences in gene expression regulate critical metabolic processes relevant for fecundity. To do so we measured several reproductive fitness parameters in $Gff^{Spi+}$ and $Gff^{Spi-}$ pregnant females and males. We began by measuring the impact of *Spiroplasma* infection on the length of tsetse's gonotrophic cycle (GC), which includes oogenesis, embryogenesis and larvigenesis. We observed that the1st, 2nd, and 3rd GCs of $Gff^{Spi+}$ females (23.0 ± 0.24, 12.6 ± 0.19, and 13.0± 0.2 days, respectively) were significantly longer than that of their age-matched counterparts that did not harbor *Spiroplasma* (21.0 ± 0.36, 11.0 ± 0.29, and 11.2± 0.33 days, respectively) (GC1, $p$ = 0.0001; GC2, $p$<0.0001; GC3, $p$ = 0.0009; log rank test) (Fig 3A and S4 Data). We next weighed pupae-stage offspring from $Gff^{Spi+}$ and $Gff^{Spi-}$ females and observed no significant difference between groups across all three GCs (GC1, $Gff^{Spi+}$ = 26.0 ± 1.4 mg, $Gff^{Spi-}$ = 24.6 ± 1.1 mg; GC2, $Gff^{Spi+}$ = 25.7 ± 1.0 mg, $Gff^{Spi-}$ = 27.4 ± 1.8 mg; GC3, $Gff^{Spi+}$ = 25.4 ± 1.7 mg, $Gff^{Spi-}$ = 26.4 ± 1.6 mg) (GC1, $p$ = 0.34; GC2, $p$ = 0.08; GC3, $p$ = 0.15; multiple t tests) (Fig 3B). These findings indicate that an infection with *Spiroplasma* does not impact pupal weight but does increase the length of tsetse's GC. As such, infection with this bacterium may result in the production of fewer offspring over the course of the female's lifespan, thus exerting a detrimental impact on the fly's overall reproductive fitness.

We have previously demonstrated that experimental depletion of *Wigglesworthia* density (via treatment with antibiotics) significantly impairs tsetse's reproduction [12,32–34]. With this in mind, we measured the relative densities of endosymbiotic *Wigglesworthia* and *Sodalis* in $Gff^{Spi+}$ and $Gff^{Spi-}$ females to determine if these symbionts are impacted by the presence of *Spiroplasma*. We determined that $Gff^{Spi+}$ and $Gff^{Spi-}$ females house similar densities of *Wigglesworthia* (Fig 3C, left graph) and *Sodalis* (Fig 3C, right graph). Because *Spiroplasma* infection did not impact the density of these tsetse symbionts, we next investigated whether the prolonged GC length presented by $Gff^{Spi+}$ females reflected a competition for resources between the bacterium and pregnant female flies. This theory is supported by the fact that in *Drosophila*, *Spiroplasma* proliferation is limited by the availability of circulating lipids [20]. The gut of 3rd instar tsetse larvae contains high levels of triacylglyceride (TAG) [35], which originate in tsetse's fat body and are transferred through hemolymph to the milk gland where they are incorporated into maternal milk secretions [36,37]. To determine if *Spiroplasma* hijacks tsetse TAG, at the nutritional expense of developing intrauterine larvae, we compared TAG levels in

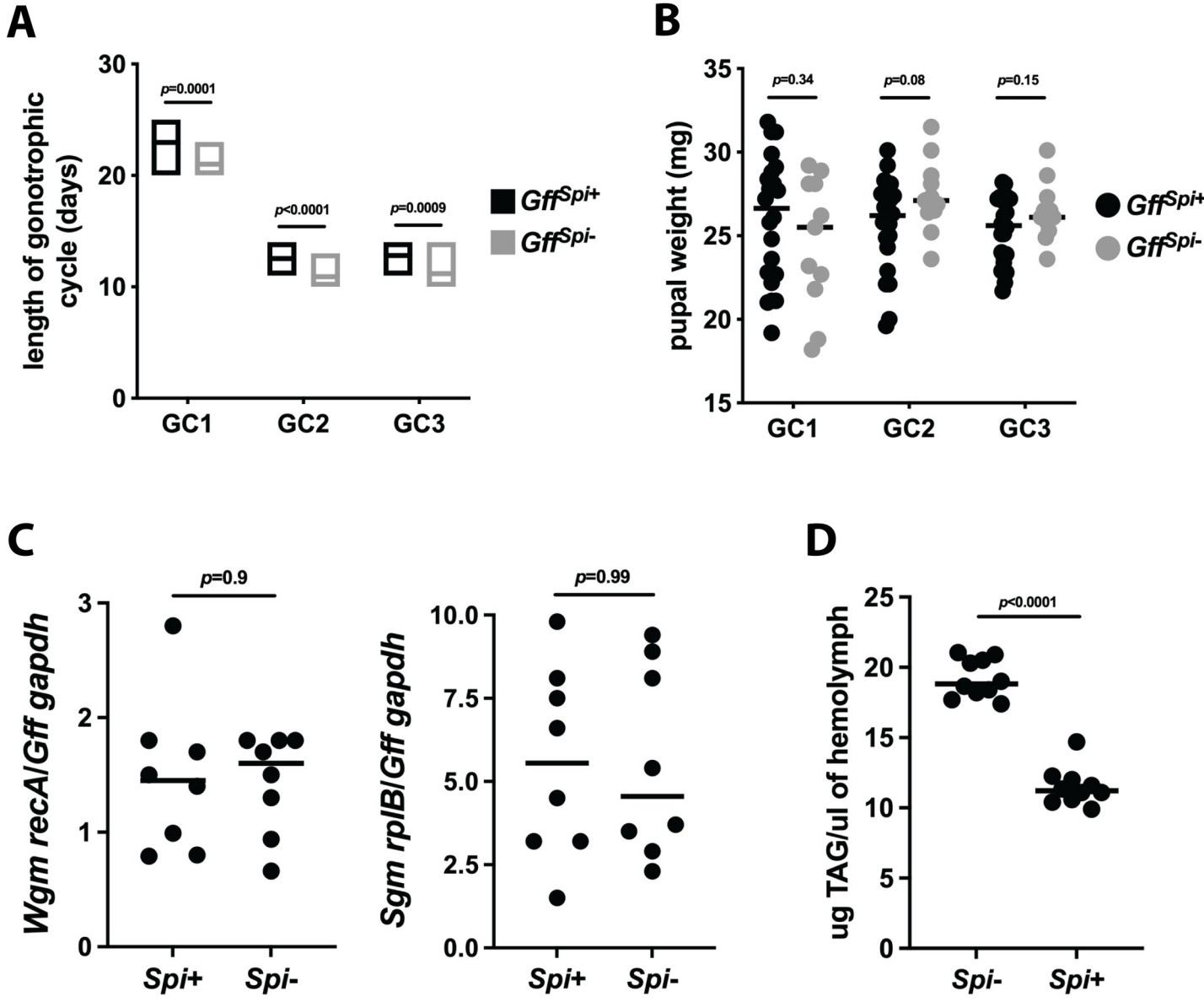

**Fig 3. Impact of *Spiroplasma* infection on the reproductive and nutritional fitness of female tsetse flies.** (A) Gonotrophic cycle (GC) length of offspring from $Gff^{Spi+}$ and $Gff^{Spi-}$ females. Age-matched, pregnant females from each group ($n$ = 34 per group) were housed in individuals cages and monitored daily to observe frequency of pupal deposition. Statistical significance was determined via log-rank test. (B) The weight of pupae deposited by $Gff^{Spi+}$ and $Gff^{Spi-}$ females. Each dot represents an individual pupa, bars represent median values of pupae form each GC. Statistical significance was determined via multiple t-tests with correction via the Holm-Šídák method. (C) Relative densities of *Wigglesworthia* and *Sodalis* in $Gff^{Spi+}$ and $Gff^{Spi-}$ females. Relative *Wigglesworthia recA and Sodalis rplB* copy number was quantified using genomic DNA derived from $Gff^{Spi+}$ or $Gff^{Spi-}$ female midguts (including the *Wigglesworthia* harboring bacteriome organ). *Wigglesworthia recA and Sodalis rplB* was normalized relative to *Gff gapdh* copy number in each sample. Each dot represents one biological replicate (three midguts per replicate), and bars indicate median values. Statistical significance was determined via students t-test. (D) Amount of triacylglyceride (TAG) circulating in the hemolymph of $Gff^{Spi+}$ and $Gff^{Spi-}$ females. Three microliters of hemolymph was extracted from two-week-old pregnant females and TAG was quantified colorimetrically via comparison to triolein standard curve (S2 Fig). Each dot represents an individual pupa, bars represent median values of pupae form each GC. Statistical significance was determined via unpaired t-test.

the hemolymph of pregnant $Spi^+$ and $Spi^-$ females. We found that pregnant $Spi^-$ females had significantly more TAG circulating in their hemolymph (19.2 ± 0.4 μg/μl) than did their age and pregnancy stage-matched $Spi^+$ counterparts (11.5 ± 0.4 μg/μl) ($p<0.0001$; t test) (Fig 3D). These findings suggest that, like in *Drosophila*, *Spiroplasma* uses tsetse lipids as an energy source. This competition between tsetse and *Spiroplasma* for dietarily limited yet metabolically

important lipids may result in the relatively long GC we observed in $Spi^+$ mothers because less of this nutrient is available for incorporation into milk secretions.

## Impact of *Spiroplasma* infection on *Gff* male reproductive fitness

Our transcriptomic data informed us that infection with *Spiroplasma* impacts the expression of several genes that encode spermatophore-associated proteins. These proteins, many of which arise from the male accessory gland and are transferred in seminal fluid, play a prominent role in sperm fitness [2,38]. We thus compared the fitness of sperm derived from $Spi^+$ compared to $Spi^-$ males to determine if infection with the bacterium impacts the fitness off *Gff* sperm. To do so we quantified the transcript abundance of *sperm-specific dynein intermediate chain* (*sdic*; VectorBase gene ID GFUI025244) in the spermathecae of 7 day old female flies two days after having mated with either $Spi^+$ or $Spi^-$ males. The *Drosophila* homologue of this gene is expressed exclusively in sperm cells [39] and has been used to quantify sperm abundance and competitiveness (as a function of motility) [40–42]. We observed that *sdic* transcript abundance in the spermathecae of females that mated with $Spi^-$ males was significantly higher than in that of their counterparts that mated with $Spi^+$ males (Fig 4). This conspicuous difference in the abundance of *sdic* transcripts expressed by sperm within the spermathecae of females that mated with $Spi^-$ versus $Spi^+$ males was surprising considering the fact that expression of this gene in the reproductive tract of field captured $Spi^-$ and $Spi^+$ males was not significantly different when measured by RNAseq (S1 Data). To validate this RNAseq data, we used RT-qPCR to measure *sdic* expression in sperm housed in the reproductive tract of 7 days old $Spi^-$ and $Spi^+$ lab reared *Gff* males (these sperm were the same age as those used to measure *sdic* transcript abundance in the spermathecae of mated females). Similar to what we observed via RNAseq, *sdic* expression was not significantly different in sperm housed in the male reproductive tract (Fig 4).

The above data indicate that *sdic* expression decreases significantly in sperm that originate from $Spi^+$ males following transfer to the female spermatheca, thus suggesting that the bacterium plays a role in either regulating the number of sperm transferred during mating and/or sperm motility (and thus competitiveness) after mating. To investigate further we quantified both the quantity and motility of sperm in the spermathecae of females 24 hrs after having mated with either $Spi^+$ or $Spi^-$ males. Manual counting indicated no significant difference in the number of stored sperm when females had mated with either $Spi^+$ (5375 ± 537 sperm per spermathecae) or $Spi^-$ (5358 ± 632 sperm per spermathecae) males (Fig 4B). We additionally quantified the abundance of stored sperm as a reflection of spermathecal fill, which measures the quantity of sperm and seminal fluid transferred from males to the spermatheca of their female mates. Male *Spiroplasma* infection status had no impact on spermathecal fill. However, when female mates were infected with *Spiroplasma* we observed a significant reduction in the size of their spermatheca compared to that of $Spi^-$ females (Kruskal-Wallis, $X^2 = 6.1763$, df = 1, $p = 0.0129$) (Fig 4C). We also investigated whether the precipitous drop in *sdic* expression in sperm transferred from $Spi^+$ compared to $Spi^-$ males was representative of a decrease in the motility of sperm derived from males in the latter group. We observed that sperm derived from $Spi^+$ males exhibited a mean beat frequency of 13.4 (±1.7) hertz (Hz), while those derived from $Spi^-$ males exhibited a mean beat frequency of 22.1 (±1.4) Hz (Fig 4D).

Taken together, these results indicate that the *Spiroplasma* infection status of male *Gff* does not impact the number of sperm they transfer during mating, but female infection status of the mating pairs does impact the number of sperm stored in their spermatheca. Additionally, *Spiroplasma* exerts a significant impact on the competitiveness of sperm stored in the spermatophore, as cells that originate from infected males are significantly less motile than are those that originate from their uninfected counterparts.

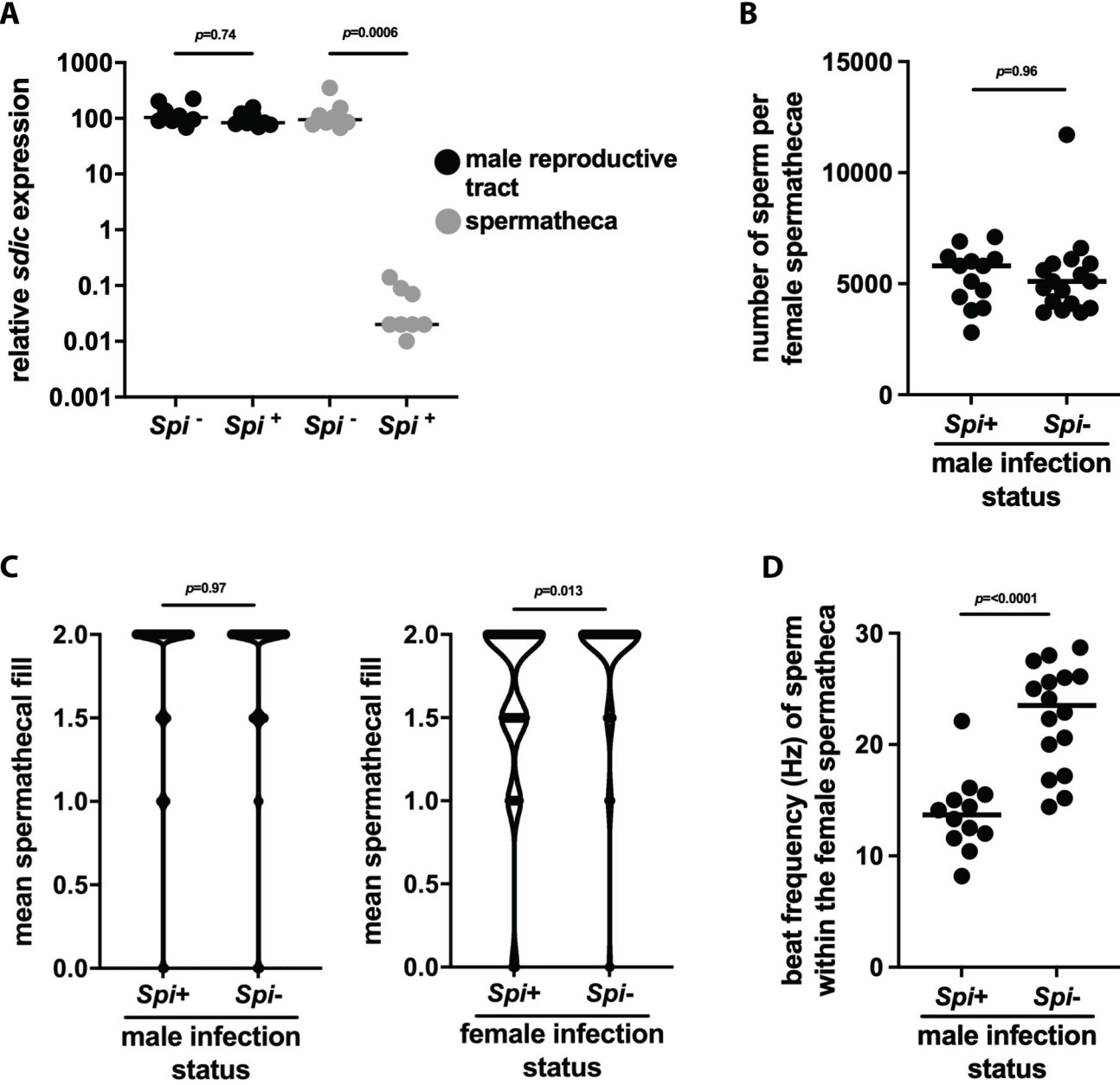

**Fig 4. The impact of *Spiroplasma* infection on male *Gff* reproductive fitness.** (A) Relative expression of *sperm-specific dynein intermediate chain* (*sdic*) in sperm located within the reproductive tract of *Gff*<sup>Spi+</sup> and *Gff*<sup>Spi-</sup> males or within the female spermatheca following copulation. *Sdic* expression in each sample was normalized relative to tsetse's constitutively expressed *pgrp-la* gene. Each dot represents one biological replicate, and bars indicate median values. Statistical significance was determined via one-way ANOVA followed by Tukey's HSD post-hoc analysis. (B) Quantification of sperm within the spermatheca of female flies that mated with either *Spi+* or *Spi-* males. Measurements were made using a Neubauer counting chamber. Each dot represents one spermatheca, and bars indicate median values. Statistical significance was determined via student's t-test. (C) Spermathecal fill of female flies that mated with either *Spi+* or *Spi-* males. Spermatheca fill data were analyzed using Kruskal-Wallis rank test using R companion and FSA software packages [75,76]. (D). Motility, as a measure of flagellar beat frequency in hertz (Hz), of sperm within the spermatheca of female flies that mated with either *Spi+* or *Spi-* males. Video recordings of sperm were acquired at a rate of 50 frames per second, and beat frequency was analyzed using FIJI and the ImageJ plugin SpermQ. Each dot on the graph represents the mean beat frequency of two sperm tails from each spermatheca. Statistical significance was determined via student's t-test.

## Vertical transmission of *Spiroplasma*

*Spiroplasma* is maternally transmitted in several insect systems, including the tsetse fly, as evidenced by the presence of the endosymbiont in the intrauterine larva [14]. However, not all tsetse populations, nor individuals within distinct populations, house the bacterium [14,15]. This heterogeneity in infection prevalence suggests that vertical transmission of the bacterium may be imperfect and thus occur at a frequency of less than 100%. We performed three distinct experiments as a means of quantitating the fidelity of *Spiroplasma* vertical transmission (see Materials and Methods for experimental details).

For experiment 1, we observed that 86% of adult offspring (from three GCs combined) derived from matings between $Spi^+$ mothers and $Spi^+$ fathers were $Spi^+$ as were 97% of adult offspring from matings between $Spi^+$ mothers and $Spi^-$ fathers. Also in experiment 1 we observed no $Spi^+$ adult offspring when mothers lacked the bacterium, regardless of their mate's infection status (Table 1 and S5 Data). For experiment 2, 67% of pupal offspring (from three GCs combined) derived from matings between $Spi^+$ mothers and $Spi^+$ fathers were $Spi^+$ as were 44% of pupal offspring from matings between $Spi^+$ mothers and $Spi^-$ fathers (Table 1 and S5 Data). In experiment 2, four matings between $Spi^-$ mothers and $Spi^+$ fathers collectively resulted in the deposition of nine pupae, four of which carried *Spiroplasma* infections. Interestingly, in three out of four mating pairs, the second offspring (GC2) was positive for the *Spiroplasma* infection while the first offspring (GC1) was negative (Table 1 and S5 Data), indicating imperfect paternal transmission at best. The varying results we obtained between these experiments could reflect the different developmental stages of the offspring we tested (experiment 1, newly eclosed adults; experiment 2, newly deposited pupae). *Spiroplasma* density decreases during development as tsetse age from larvae through pupation and eclosion to adulthood [14]. Hence, it remains to be seen whether the *Spiroplasma* present in the pupae analyzed in experiment 2 would persist through the lengthy pupal stage (~ 30 days) and into adulthood.

Finally, in experiment 3, we determined *Spiroplasma* infection status of mothers only. Of the $Spi^+$ mothers observed, 73% (11/15) and 100% (9/9) of their pupal offspring from GCs 1 and 2, respectively, were infected with *Spiroplasma* (no $Spi^+$ mothers in this experiment survived long enough to produce GC 3 offspring). Nine out of the 22 total progeny from $Spi^-$ mothers were $Spi^+$, although transmission did not occur across sequential GCs (S5 Table in

**Table 1. Vertical transmission of *Spiroplasma* across three gonotrophic cycles (GCs).**

| | *Spiroplasma* infection status | | | # of *Spiroplasma* infected offspring/total offspring sample/GC | | | |
|---|---|---|---|---|---|---|---|
| experiment #[a] | mother | father | number of mating pairs[b] | GC1 | GC2 | GC3 | total % transmission[c] |
| Exp 1 | + | + | 7 | 6/7 | 6/7 | 6/7 | 86% (18/21) |
| | | - | 13 | 13/13 | 13/13 | 12/13 | 97% (38/39) |
| | - | + | 9 | 0/9 | 0/9 | 0/9 | 0% (0/27) |
| | | - | 5 | 0/5 | 0/5 | 0/5 | 0% (0/15) |
| Exp 2 | + | + | 5 | 4/5 | 2/3 | 0/1 | 67% (6/9) |
| | | - | 7 | 6/7 | 3/5 | 2/2 | 79% (11/14) |
| | - | + | 4 | 1/4 | 3/3 | 0/2 | 44% (4/9) |
| | | - | 1 | 0/1 | 0/1 | 0/0 | 0% (0/2) |

[a]In experiment 1 *Spiroplasma* infection prevelance was determined using adult offspring, while in experiment 2 *Spiroplasma* infection prevelance was determined using pupal offspring.

[b]Number of mating pairs at the beginning of each experiment. Not all mothers survived for three GCs.

[c]Percentage of *Spiroplasma* infected offspring derived from each parental mating pair over the course of all three GCs combined.

S1 Appendix and S5 Data). While this experiment suggests possible paternal *Spiroplasma* transmission, we cannot confirm the route of infection because we did not test the *Spiroplasma* status of these fathers.

Taken together, these transmission data indicate that *Spiroplasma* is vertically transmitted via maternal lineages with high frequency, although paternal transmission may provide a less frequent route.

## Discussion

Reproductive tissue-associated heritable endosymbionts affect their arthropod host's physiology to facilitate their transmission. In tsetse, both natural populations and laboratory *Gff* lines are found to house heterogenous infections with parasitic *Spiroplasma*, but little is known about the physiological impact(s) of this microbe on tsetse reproduction. Our transcriptomic analyses revealed that *Spiroplasma* infection significantly alters gene expression in reproductive tissues of both male and female *Gff*. In particular, amongst the genes impacted by *Spiroplasma* infection in females, significantly more female-biased genes are up-regulated, and male-biased genes are down-regulated when compared to their uninfected counterparts. Using a laboratory line of *Gff* that carries a heterogenous infection with *Spiroplasma*, we discovered that infection with the bacterium results in a reduction in fecundity as evidenced by a significantly longer gonotrophic cycle in infected females compared to their uninfected counterparts. Loss of fecundity likely results from the decreased levels of hemolymph lipids in *Spiroplasma* infected females, which suggests that the bacterium competes with its host during pregnancy for nutrients that are critical for larval development. Additionally, we observed a dramatic reduction in the abundance of a sperm-specific *sdic* transcripts following transfer of spermatozoa from *Spi*⁺ males to their mate's spermatheca. Upon further investigation we determined that these sperm cells exhibit compromised motility and thus likely decrease competitiveness. Finally, our *Spiroplasma* transmission studies suggest maternal transmission of the bacterium occurs with high fidelity from infected mothers to each of their offspring. However, we also observed evidence of paternal transmission of the bacterium, which could explain the heterogenous infections we observed in this laboratory line. Collectively, our findings significantly enhanced fundamental knowledge on the tsetse-*Spiroplasma* symbiosis with implications for other arthropods. Additionally, the information obtained in this study may be applicable to the development of novel tsetse control strategies for population reduction.

Sex-biased gene expression strongly influences sexual phenotypes in most animals [27,28], and a large proportion of sex-biased genes are expressed in reproductive tissues [43,44]. With this in mind, the differences in sex-biased gene expression we observed in between $Gff^{Spi-}$ and $Gff^{Spi+}$ females suggest that housing the bacterium could induce beneficial reproductive phenotypes. Our transcriptomic data from $Gff^{Spi+}$ females showed that female-biased genes are up-regulated while male-biased genes are down-regulated relative to their uninfected ($Gff^{Spi-}$) counterparts. Notably, we observed that *Spiroplasma* infection induced a significant upregulation of genes that encode tsetse milk gland proteins (MGPs). These tsetse specific molecules are abundantly expressed in the accessory gland of pregnant females and are a prominent component of tsetse milk [1,45]. MGPs are thought to be mostly lipid carriers, and experimental evidence suggests that they act in this capacity as lipid emulsification agents and possible phosphate carrier molecules [37]. Thus, when faced with depleted levels of circulating TAG, which comprises a substantial fraction of the lipids found in tsetse milk [35], *Spiroplasma* infected females may adaptively increase MGP production in an effort to scavenge and transport other lipids to developing intrauterine larvae. This outcome may reflect one *Spiroplasma* induced change in sex-biased gene expression that confers a fitness benefit to $Gff^{Spi+}$ females.

The depletion of circulating TAG by hemolymph borne *Spiroplasma* likely accounts for the increase we observed in GC length in pregnant mothers that harbor the bacterium. Despite the longer larval development period we observed for progeny of *Spi*+ compared to *Spi*- mothers, the weight of pupal offspring deposited was similar between the two groups. Because female tsetse produce unusually few offspring (6–8) over the course of their lifespan (compared to other insects), an increase in GC length would likely result in a significant reduction in population size over time. In fact, infection with a trypanosome strain that induces a metabolically costly immune response also lengthens tsetse's GC by a duration similar to that (approximately 2 days) induced by infection with *Spiroplasma* [46]. Mathematical modelling indicates that this increase in GC length would theoretically reduce tsetse fecundity by approximately 30% over the course of a female's reproductive lifespan [46]. Thus, a moderate trypanosome infection prevalence of 26%, which is similar to the *Spiroplasma* infection prevalence in field captured *Gff* (5–34%, depending on population geographic location) [14,15], could significantly decrease fly population size [46]. The models also predict that infection prevalence above these frequencies would result in a population crash.

Our results indicate that infection with *Spiroplasma* also exerts a significant impact on reproductive processes in laboratory reared *Gff* males. We observed that sperm from both *Spi*+ and *Spi*- males expressed the same abundance of sperm-specific *sdic* transcripts prior to mating, but that following insemination and transfer to the female spermatheca, sperm from *Spi*+ males expressed conspicuously fewer transcripts of this gene than did sperm that had originated from *Spi*- males. The *Drosophila* homolog of this gene encodes a cytoplasmic dynein intermediate chain that is necessary for the proper function of the cytoplasmic dynein motor protein complex [39], which is involved in sperm motility [47]. Accordingly, one prominent phenotype we observed was that following insemination and transfer to the female spermatheca, the motility of sperm that had originated from *Spi*+ males was impaired when compared to that of sperm that had originated from *Spi*- males. This finding implies that *sdic* exhibits the same function in *Gff* as it does in *Drosophila*. Motility is of paramount importance to sperm competitiveness [48], and sperm with impaired motility have less fertilization success than heathy sperm [49]. Field data indicate that wild populations of *Gff* females can exhibit polyandry and store sperm from more than one mate [7]. Our data suggest that sperm transferred by *Spi*+ males may be at a competitive disadvantage in comparison to sperm derived from *Spi*- males because they exhibit reduced motility. It remains to be determined if, as a means of promoting polyandry, females seek additional mates if they first copulate with a *Spi*+ male that transfers motility compromised sperm.

*Spiroplasma* mediated mechanisms that influence the regulation of *sdic* expression are currently unknown. *Spiroplasma* could mark sperm in the reproductive tract of *Gff* males [50], possibly via post-transcriptional modification of *sdic*, such that sperm motility is compromised following insemination and transfer to the female spermatheca. Alternatively, female contributions to the spermatophore could also influence *sdic* expression after sperm are packaged into the structure. In *Spiroplasma* infected females, we have observed 12 DE genes that encode products identified as constituents spermatophore. Further experiments are necessary to determine if any of these *Spiroplasma* modified female products, including two transcription factors and three serine protease inhibitors, play a role in the sperm modifications we observed here.

Infections with heritable endosymbionts can come at a significant metabolic cost to their host [51]. This outcome appears to also be the case with the *Spiroplasma* and tsetse symbiosis, and results in a significant reduction in the reproductive fitness of both female and male flies. However, from an evolutionary perspective, these microbes, including *Spiroplasma*, usually also provide an overall fitness advantage in order for infections to be maintained. In the tsetse

model system, the fly, the commensal symbiont *Sodalis*, as well as pathogenic trypanosomes, typically compete for host nutrients as they are auxotrophic for metabolically critical B vitamins [12,52–54]. Interestingly, these nutrients are present in low quantities in tsetse's vertebrate blood specific diet, and they are instead supplemented via the mutualistic endosymbiont, *Wigglesworthia* [11–13]. Despite the negative impacts of *Spiroplasma* on tsetse's fecundity, this bacterium also benefits its host by creating an environment within the fly that is hostile to parasitic trypanosomes [15]. The parasite resistance phenotype conferred by *Spiroplasma* infections could arise from induced host immune responses, or could reflect competition for critical nutrients (i.e., found in the bloodmeal and/or produced by tsetse) that both organisms require to sustain their metabolic needs [55–57]. This effect of inhibiting metabolically costly trypanosome infections could thus offset the cost of housing similarly detrimental *Spiroplasma* infections. Additionally, we demonstrate that while *Spiroplasma* can be both matrilineally and patrilineally transmitted, vertical transmission of the bacterium within our *Gff* colony does not occur at a frequency of 100%. This data corroborates observations from the field, which reveal that individual flies between and within distinct populations also present heterogeneous *Spiroplasma* infections [14,15]. In our study we observed a high frequency of maternal transmission into pupal and adult offspring as well as low frequency paternal transmission into pupal offspring. If *Spiroplasma* detected in the pupal stage survive and colonize adults, this outcome could contribute to the heterogenous infection prevalence we observe in the *Gff* lab line. The absence of steadfast vertical transmission of *Spiroplasma* by *Gff* could facilitate fly survival by reducing the population level detrimental impact (i.e., prolonged GC length and reduced sperm motility) associated with housing the bacterium. This may represent another advantageous factor that facilitates *Spiroplasma's* ability to sustain infections within the fly across multiple generations.

*Gff* is the prominent vector of pathogenic African trypanosomes in east and central Africa, and reducing the size of fly populations is currently the most effective method for controlling parasite transmission. One means of accomplishing this is through the use of sterile insect technique (SIT), which involves the sequential release of a large number of sterilized males into a target environment [58,59]. Because of their large number, these infertile males outcompete wild males for resident female mates, and the population size drops significantly, or the population is eliminated [60,61]. Knowledge gained from this study, in conjunction with that from previous studies, will have a significant impact on tsetse rearing efficiency for the SIT programs. Specifically, decreased fecundity can have a negative impact on the success of colony maintenance. However, enhanced resistance to trypanosomes is advantageous for SIT applications as the released males, which also feed exclusively on vertebrate blood, have the potential to serve as vectors of disease. Thus, *Spiroplasma* has the potential to reduce the impact of this shortcoming by inducing a trypanosome refractory phenotype in released males such that they would be less likely vector disease, thus increasing the overall safety and efficacy of the control program.

## Materials and methods

### Field sampling

*Glossina fuscipes fuscipes* (*Gff*) were collected from the Albert Nile river drainage in Northwest Uganda. Sampling sites included three villages from the Amuru district: Gorodona (GOR; 3˚ 15'57.6"N, 32˚12'28.8"E), Okidi (OKS; 3˚15'36.0"N, 32˚13'26.4"E), and Toloyang (TOL; 3˚ 15'25.2"N, 32˚13'08.4"E). Flies were sampled using biconical traps and wing fray data were recorded (stage 2 for the majority of samples, stage 3 for a small subset of samples, corresponding approximately to 3 and 4 week-old adults, respectively). All females analyzed had been

mated based on the visual presence of intrauterine larva or a developing oocyte in their ovaries. The *Spiroplasma* infection prevalence in flies from the Goronda and Okidi regions was about 46% [15]. Reproductive tissues (may have been contaminated with milk gland tissue, which is composed of an extensive tubular network that is difficult to separate from female reproductive tissues) from all flies were dissected in the field in sterile 1x phosphate buffered saline (PBS) and flash frozen in liquid nitrogen for later RNA extraction.

## Data generation (Illumina HiSeq), transcriptome assembly and quality assessment

Total RNA from *Gff* female (ovaries, spermathecae, uterus and oocysts) and male (testes, accessory glands and ejaculatory duct) reproductive tracts was extracted according to TRIzol reagent manufacturer's instructions (Thermo Fisher Scientific, USA). Total RNA was treated with Ambion TURBO DNA-free DNase (Thermo Fisher Scientific, USA) and RNA quality was evaluated using an Agilent 2100 Bioanalyzer. An aliquot from each RNA was reverse transcribed into cDNA and screened for *Spiroplasma* using PCR amplification assay with symbiont-specific primers as described [15]. Prior to pooling RNA for sequencing (see below), the same cDNAs were used to confirm the sex of the field collected tissues by performing PCR using primers that amplify sex specific genes (see S4 Table in S1 Appendix for gene names and amplification primers). RNA from three female or male reproductive tracks were pooled into distinct biological replicates for RNA-seq analysis. We analyzed two biological replicates from $Gff^{Spi+}$ females and three biological replicates from $Gff^{Spi+}$ males, and three biological replicates from $Gff^{Spi-}$ females and two biological replicates from $Gff^{Spi-}$ males. Ribosomal reduction libraries were prepared with the Ribo-Zero Gold rRNA removal kit (human/mouse/rat) MRZG12324 (Illumina, USA) and sequenced at the Yale University Center of Genome Analysis (YCGA) on an Illumina HiSeq 2500 machine (75bp paired-end reads). In order to obtain information on *Spiroplasma* transcripts as well as the tsetse host, we chose Ribosomal Reduction libraries over *m*RNA (poly A) libraries.

## RNA-seq data analysis

RNA-seq sequencing produced an average of 96 million reads across all biological samples. Reads were mapped to the *Gff* reference genome (accession # JACGUE000000000 from NCBI BioProject PRJNA596165) using HISAT2 v2.1.0 with default parameters [62,63]. The annotations for the assembly were ported over from the annotation of the previous assembly (https://vectorbase.org/vectorbase/app/) using the UCSC liftOver software suite of programs [64]. The function 'htseq-count' in HTSeq v0.11.2 [65] was then used to count the number of reads aligned to the annotated genes in the reference genome with the option "-s reverse" because we generated strand-specific forward reads in RNA-seq libraries. We evaluated the number of DE genes between groups of female and male reproductive organs from $Gff^{Spi+}$ and $Gff^{Spi-}$ using the HTSeq output as input into DESeq2 v1.22.1 [66]. We first developed a DESeq2 model to predict differential gene expression as a function of sex and *Spiroplasma*-infection status, and the interaction between sex and *Spiroplasma*-infection status. We then extracted the log$_2$ fold changes (log$_2$FCs) for each gene with false discovery rate (FDR) adjusted *p*-values [24]. Finally, we defined genes as DE if the log$_2$ fold change (log$_2$FC) was significantly different from 0, and we only considered genes with adjusted *p*-values as expressed for downstream analysis.

We next conducted a PC and HC analysis to analyze normalized expression count data from the DESeq2 program. We used a regularized log transformation of the normalized data created by "rlog" function DESeq2 [66]. For the HC by sample-to-sample distance, we used

"pheatmap" function in R. We measured the expression abundance as the sum of TPM that were calculated using the TPM Calculator [67].

Because the function of genes in *Gff are* not well annotated, we performed 'tblastx' in CLC Genomics Workbench (CLC Bio, Cambridge, MA) with *Gff* transcript to search for predicted protein sequences bearing the closest homology to those from *G. m. morsitans*, *Musca domestica*, and *Drosophila melanogaster*. Using gene-associated GO terms [31], we performed enrichment analysis of gene ontology (GO) terms with the topGO R package [68].

## Laboratory maintenance of *Gff* and determination of *Spiroplasma* infection status

*Gff* pupae were obtained from Joint FAO/IAEA IPCL insectary in Seibersdorf, Austria and reared at Yale University insectary at 26˚C with 70–80% relative humidity and a 12 hr light: dark photo phase. Flies received defibrinated bovine blood every 48 hr through an artificial membrane feeding system [69].

*Spiroplasma* infection status of all lab-reared female, male and pupal *Gff* was determined by extracting genomic DNA (gDNA) from individual whole organisms or fly legs using a DNeasy Blood and Tissue kit according to the manufacturer's (Qiagen) protocol. To confirm that intact gDNA was successfully extracted, all samples were subjected to PCR analysis using primers that specifically amplify *Gff tubulin* or microsatellite region *GpCAG* (as controls for genomic DNA quality) and *Spiroplasma 16S rRNA*. All PCR primers used in this study are listed in S4 Table in S1 Appendix. The *Spiroplasma 16S rRNA* locus was amplified by PCR (as described in [14]) using the following parameters: initial denaturation at 94˚C for 5 min; 34 cycles of 94˚C for 45 s, 59˚C for 45 s, and 72˚C for 1 min; and a final extension at 72˚C for 10 min. *Spiroplasma* PCR reactions were carried out in a volume of 25 μl containing 1.5 μl gDNA. PCR products were analyzed on 2% agarose gels, and samples were considered infected with *Spiroplasma* if the expected PCR product of 455 bp was detected.

To confirm the sensitivity of this leg snip PCR assay, and thus eliminate the possibly of false negative outcomes, we randomly selected 10 *Spiroplasma* negative samples and two positive controls (as determined by endpoint PCR described above) and subjected them RT-qPCR using the *Spiroplasma ropB*. All RT-qPCR results were normalized to tsetse's constitutively expressed *pgrp-la* gene (S2 Fig). RT-qPCR confirmed our endpoint PCR data (RT-qPCR output shown in S6 Data).

## Impact of *Spiroplasma* infection on *Gff* metabolism and reproductive fitness

*Hemolymph triacylglyceride (TAG) assay*: Hemolymph (3 μl/fly) was collected (as described in [70]) from two-week-old pregnant female *Gff*, centrifuged (4˚C, 3000xg for 5 minutes) to remove bacterial cells, diluted 1:10 in PBS containing 1.2 μl/ml of 0.2% phenylthiourea (to prevent hemolymph coagulation) and immediately flash frozen in liquid nitrogen. Hemolymph TAG levels were quantified colorimetrically by heating samples to 70˚C for 5 min followed by a 10 min centrifugation 16,000xg. Five μl of supernatant was added to 100 μl of Infinity Triglycerides Reagent (Thermo Scientific) and samples were incubated at 37C for 10 min. Absorbance was measured at 540nm using a BioTek Synergy HT plate reader [71]. All *Gff* sample spectra data were compared to that generated from a triolein standard curve (0–50 μg, 10 μg increments; S3 Fig).

## Impact on fecundity and endosymbiont density

The effect of *Spiroplasma* infection on *Gff* fecundity was measured by quantifying the length of three gonotrophic cycles (GC) and by weighing pupal offspring. To measure GC length, *Gff*

females were mated as five days old adults and thereafter maintained in individual cages. All females were monitored daily to determine when they deposited larvae, and all deposited larvae were weighed.

To confirm the sensitivity of this leg snip PCR assay, and thus eliminate the possibly of false negative outcomes, we randomly selected 10 *Spiroplasma* negative samples and two positive controls (as determined by endpoint PCR described above) and subjected them RT-qPCR using the *Spiroplasma ropB*. All RT-qPCR results were normalized to tsetse's constitutively expressed *pgrp-la* gene (S2 Fig). RT-qPCR confirmed our endpoint PCR data (RT-qPCR output shown in S6 Data).

## Sperm-specific gene expression

Individual five-day old male and female flies (each fed twice) were placed together into tubular cages (height, 12.7 cm; diameter 6 cm) and allowed to mate for two days. Individual mating pairs were subsequently separated, and RNA (using Trizol reagent) was isolated from male and female reproductive tracts. RNA was then DNase treated and reverse transcribed into cDNA (using a Bio-Rad iScript cDNA synthesis kit) by priming the reaction with random hexamers. All males were then screened by PCR as described above to determine their *Spiroplasma* infection status.

*Gff sperm-specific dynein intermediate chain* (*sdic*; GFUI025244) transcript abundance (measured via RT-qPCR, primers used are listed in S4 Table in S1 Appendix) was used as a proxy measurement of sperm density and sperm fitness in male reproductive tracts and the spermathecae of pregnant females [39,47]. All RT-qPCR results were normalized to tsetse's constitutively expressed *pgrp-la* gene, and relative expression of *sdic* was compared between male reproductive tracts and sperm (in the female spermathecae) that originated from $Spi^+$ and $Spi^-$ males. All RT-qPCR assays were carried out in duplicate, and replicates quantities are indicated as data points on corresponding figures. Negative controls were included in all amplification reactions.

## Sperm quantification and spermathecal fill assays

Sperm abundance in the spermathecae of females that had mated with either $Spi^+$ and $Spi^-$ males was quantified by both direct cell counting and by measuring spermathecal fill. For direct counting, spermathecae were excised from mated females 24 hrs post-copulation with either $Spi^+$ or $Spi^-$ males and placed into 10 μl of HEPES-buffered saline solution [145 mM NaCl, 4 mM KCl, 1 mM MgCl2, 1.3 mM CaCl2, 5 mM D-glucose, 10 mM 4-(2-hydroxyethyl)-1-piperazineethane- sulfonic acid (HEPES), pH 7.4] in the well of a concave glass microscope slide. Spermathecae were gently poked with a fine needle and sperm were allowed to exude from the organ for 5 minutes. Samples were subsequently diluted 1:100 in HEPES-buffered saline and counted using a Neubauer counting chamber. To measure spermathecal fill, spermathecae from mated females were microscopically dissected 24h post-copulation in physiological saline solution (0.9% NaCl) and assessed subjectively at 100x magnification. Spermathecal fill of each individual organ was scored to the nearest quarter as empty (0), partially full (0.25, 0.50, or 0.75), or full (1.0) (S4 Fig), and the amount of sperm transferred was then computed as the mean spermathecal filling values of the spermathecae pairs [72,73].

## Sperm motility assays

Spermathecae were excised from females 24 hrs post-copulation with either $Spi^+$ and $Spi^-$ males and placed into 10 μl of HEPES-buffered saline solution [145 mM NaCl, 4 mM KCl, 1 mM MgCl2, 1.3 mM CaCl2, 5 mM D-glucose, 10 mM 4-(2-hydroxyethyl)-1-piperazineethane-

sulfonic acid (HEPES), pH 7.4] in the well of a concave glass microscope slide. Spermathecae were gently poked with a fine needle and sperm were allowed to exude from the organ for 5 minutes. Sperm beating was recorded using an inverted microscope (10x phase contrast, Zeiss Primovert) that housed a charge-coupled camera (Zeiss Axiocam ERc 5s). Two sperm tails per sample were analyzed. Recordings were acquired at a rate of 30 frames per second, and beat frequency was analyzed using FIJI and the ImageJ plugin SpermQ [74].

## Vertical transmission efficiency

Three separate experiments, performed by different researchers in different insectaries, were implemented to monitor the dynamics of *Spiroplasma* transmission.

*Experiment 1*: Individual five-day old male and female flies (each fed twice) were placed together into small tubular cages (height, 12.7 cm; diameter 6.0 cm) and allowed to mate for two days. Males were subsequently removed from the cages (and frozen for future analysis) and pregnant females were fed every other day throughout the course of three GCs. Pupae were collected from each female and housed separately until adult emergence. Genomic DNA from legs of parent flies and all offspring was purified and subjected to PCR analysis to determine *Spiroplasma* infection status, as indicated above in subsection '*Laboratory maintenance of Gff and determination of Spiroplasma infection status*'.

*Experiment 2*: Forty 7–9 day old male *Gff* were released into large mating cages (45x45x45 cm), and 15 minutes later an equal number of 3–5 day old females were introduced. Mating was observed under standard rearing conditions. As soon as copulation began, mating pairs were collected and placed into individual cages (4 cm diameter x 6 cm height) where they were kept together for 24 h. Males were subsequently removed (and conserved in absolute ethanol at -20˚C for further analysis), and mated females were pooled together in a larger cage and maintained under normal rearing conditions for 10 days. Finally, pregnant females were again separated into individual cages and maintained under normal conditions over the course of three GCs (~40 days). Individual pupa were collected from each female following each GC and conserved in ethanol at -20˚C for further analysis 24h and 72 h post deposition.

*Experiment 3*: This experiment was performed the same in the same manner as was experiment 2, with the exception that male flies were disposed of after mating.

## Statistical analyses

All statistical analyses were carried out using GraphPad Prism (v.9), Microsoft Excel or RStudio (v.1.2.5033 and v.1.3.1073). All statistical tests used, and statistical significance between treatments, and treatments and controls, are indicated on the figures or in their corresponding legends. All sample sizes are provided in corresponding figure legends or are indicated graphically as points on dot plots. Biological replication implies distinct groups of flies were collected on different days.

## Supporting information

**S1 Appendix. Supporting tables S1-S5.**
(DOCX)

**S1 Fig. Distribution of differentially expressed transcript products in functional classes analyzed by Gene Ontology (GO) enrichment analysis.** The bar diagrams show the significantly enriched GO terms among **(A)** the up-regulated and **(B)** the down-regulated genes in *Gff*^*Spi+*^ males, and among **(C)** the up-regulated and **(D)** the down-regulated genes in *Gff*^*Spi+*^ females. The number of genes associated with the corresponding GO terms to the number of

genes belonging to that GO term within the entire set of genes in the genome is shown for each bar. The colors associated with the different bars denote the two different GO categories; BP: Biological Process, MF: Molecular Function.
(TIF)

**S2 Fig. Relative expression of *peptidoglycan recognition protein la* (*pgrp-la*) in the reproductive tract of females that mated either *Gff^Spi+* or *Gff^Spi-* males.** *Pgrp-la* expression in each sample was normalized relative to geometrical mean of tsetse's constitutively expressed *gapdh and β-tubulin* genes. Each dot represents one biological replicate, and bars indicate median values. Statistical significance was determined via students t-test.
(TIF)

**S3 Fig. Triolein standard curve used to determine the concentration of triacylglyceride circulating in the hemolymph of pregnant *Gff^Spi+* compared to *Gff^Spi-* females.** 0–50 μg aliquots of triolein were mixed with 100 μl of Infinity Triglycerides Reagent (Thermo Scientific) and samples were incubated at 37C for 10 min. Absorbance was measured at 540nm using a BioTek Synergy HT plate reader.
(TIF)

**S4 Fig. Diagrammatic guide used to visually quantify spermathecal fill in females 24 h post-copulation.** Image generated by Dr. Güler Demirbas-Uzel.
(TIF)

**S1 Data. Gene expression data created in DESeq2 analysis, homologies in other Diptera of DE genes, and DE genes that constitute spermatophore proteins.**
(XLSX)

**S2 Data. Transcripts Per Million (TPM) for each sample.**
(XLSX)

**S3 Data. GO enrichment analysis results.**
(XLSX)

**S4 Data. Gonotrophic cycle (GC) length in relation to maternal *Spiroplasma* infection status.**
(XLSX)

**S5 Data. *Spiroplasma* vertical transmission data.**
(XLSX)

**S6 Data. RT-qPCR confirmation of *Spiroplasma* infection status of samples used for vertical transmission experiments.** Ten *Spiroplasma* negative samples and two *Spiroplasma* positive samples (as determined by endpoint PCR described in the Materials and Methods) were randomly selected for this analysis (describe in the Materials and Methods, subsection '*Laboratory maintenance of Gff and determination of Spiroplasma infection status*').
(XLSX)

## Acknowledgments

We thank Dr. Güler Demirbas-Uzel (Insect Pest Control Laboratory, Joint FAO/IAEA Programme of Nuclear Techniques in Food & Agriculture, Vienna, Austria) for technical assistance with spermathecal fill assays. We are thankful for support from the United Nations, International Atomic Energy Association sponsored Coordinated Research Project entitled

'Improvement of Colony Management in Insect Mass-Rearing for SIT Applications'. We thank Dr. Peter Takac, at the Slovakian Academy of Sciences, for supplying tsetse pupae.

## Author Contributions

**Conceptualization:** Jae Hak Son, Brian L. Weiss, Daniela I. Schneider, Kiswend-sida M. Dera, Fabian Gstöttenmayer, Geoffrey M. Attardo, Maria Onyango, Adly M. M. Abd-Alla, Serap Aksoy.

**Data curation:** Jae Hak Son, Brian L. Weiss, Daniela I. Schneider, Kiswend-sida M. Dera, Fabian Gstöttenmayer, Geoffrey M. Attardo, Maria Onyango, Adly M. M. Abd-Alla, Serap Aksoy.

**Formal analysis:** Jae Hak Son, Brian L. Weiss, Daniela I. Schneider, Kiswend-sida M. Dera, Fabian Gstöttenmayer, Geoffrey M. Attardo, Maria Onyango, Adly M. M. Abd-Alla, Serap Aksoy.

**Funding acquisition:** Serap Aksoy.

**Investigation:** Jae Hak Son, Brian L. Weiss, Daniela I. Schneider, Kiswend-sida M. Dera, Fabian Gstöttenmayer, Robert Opiro, Richard Echodu, Norah P. Saarman, Geoffrey M. Attardo, Maria Onyango, Adly M. M. Abd-Alla, Serap Aksoy.

**Methodology:** Jae Hak Son, Brian L. Weiss, Daniela I. Schneider, Kiswend-sida M. Dera, Fabian Gstöttenmayer, Geoffrey M. Attardo, Adly M. M. Abd-Alla, Serap Aksoy.

**Project administration:** Brian L. Weiss, Serap Aksoy.

**Resources:** Brian L. Weiss, Robert Opiro, Richard Echodu, Norah P. Saarman, Adly M. M. Abd-Alla, Serap Aksoy.

**Software:** Jae Hak Son, Brian L. Weiss.

**Supervision:** Brian L. Weiss, Adly M. M. Abd-Alla, Serap Aksoy.

**Validation:** Jae Hak Son, Brian L. Weiss.

**Visualization:** Jae Hak Son, Brian L. Weiss.

**Writing – original draft:** Jae Hak Son, Brian L. Weiss, Adly M. M. Abd-Alla, Serap Aksoy.

**Writing – review & editing:** Jae Hak Son, Brian L. Weiss, Adly M. M. Abd-Alla, Serap Aksoy.

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
