## [Decision Letter · Decision Letter 0]

14 Jun 2021

Dear Dr. Weiss,

Thank you very much for submitting your manuscript "Infection with endosymbiotic Spiroplasma disrupts tsetse (Glossina fuscipes fuscipes) metabolic and reproductive homeostasis" for consideration at PLOS Pathogens. As with all papers reviewed by the journal, your manuscript was reviewed by members of the editorial board and by several independent reviewers. In light of the reviews (below this email), we would like to invite the resubmission of a significantly-revised version that takes into account the reviewers' comments.

Three reviewers have major requests before the manuscript can be considered for submission. I would recommend to address all of them and submit a revised version.

We cannot make any decision about publication until we have seen the revised manuscript and your response to the reviewers' comments. Your revised manuscript is also likely to be sent to reviewers for further evaluation.

Sincerely,

Bruno Lemaitre

Guest Editor

PLOS Pathogens

Kirk Deitsch

Section Editor

PLOS Pathogens

Kasturi Haldar

Editor-in-Chief

PLOS Pathogens

orcid.org/0000-0001-5065-158X

Michael Malim

Editor-in-Chief

PLOS Pathogens

orcid.org/0000-0002-7699-2064

Three reviewers have major requests before the manuscript can be considered for submission. I would recommend to address all of them and submit a revised version.

Reviewer's Responses to Questions

**Part I - Summary**

Reviewer #1: This manuscript first describes the effect of Spiroplasma infection on tsetse fly reproductive tract transcriptome. Authors then performed functional experiments to assess the impact of the infection on both male and female reproductive ability. They conclude about a negative impact of Spiroplasma infection on both female and male reproductive fitness.

This work is overall very insightful and makes a significant contribution to the field, and the manuscript itself is well written.

Reviewer #2: This manuscript analyses the interaction between Spiroplasma and tse tse flies. There are three components - RNAseq comparison of S+ and S- individuals, examination of some phenotype associations, notably life history and sperm mobility, and transmission patterns. All elements are certainly novel and given the importance of tse tse, timely and important.

The impact on sperm motility was interesting and a phenotype not seen before in symbiosis studies. The paternal transmission is also very interesting (though I have some technical caveats to clear up).

Potential weaknesses of the study:

-I found the link between the RNAseq observations and the experiments a little forced. The experiments aren't really testing strong predictions from the RNAseq in my mind. Thus, whilst I found all the components interesting, I wasn't convinced by the synthesis.

-the RNAseq material is from the field and is limited in sample size within each sex; this raises the potential for confounding influences between S+ and S- individuals, which is particularly important in this study, as the most interesting comparisons are S+ vs S- within sex.

-For the transmission (and other) studies- Spiroplasma infection status is ascertained from endpoint PCR. Previous work has shown quite high variability in Spiroplasma titre in tse tse- potentially making more sensitive qPCR important for diagnosis of infection. This is important in RNAseq experiments and critical in transmission experiments, where one might diagnose paternal transmission in cases where the mother in fact has a low titre infection. Within this, I'd also like to know if leg snips have been validated as a means of diagnosing Spiroplasma status - in particular is there a false negative rate.

-Also in the transmission measures - L416 Technically, if you have paternal and maternal transmission, you can only work out maternal vertical transmission rates in cases where the female is infected and the male is known to be uninfected, and in most cases, looking at the supplementary material data, the male isn't directly tested in these experiments. Thus, the experiments may overscore maternal inheritance.

Reviewer #3: This paper is centered on Glossina fuscipes fuscipes a tsetse species, which is a prominent vector of human African trypanosomiasis. The aim is to analyze and characterize the effects of the Spiroplasma endosymbiont on the reproductive biology at the physiological and metabolic levels. This aim is strongly based on previously acquired knowledge: 1) Spiroplasma, in Drosophila females affects fecundity as a a consequence of nutritional competition between the fly and bacterium for metabolically important lipids. 2) Populations of G. fuscipes are polymorphic for the presence of this symbiont. 3) Spiroplasma enhances refractoriness to trypanosomes, but the mechanism is unknown. It follows that the knowledge acquired from this study will be of extreme interest both for understanding the biological role of the presence of Spiroplasma in G. fuscipes and eventually for exploiting this symbiont for applicative aspects related to the control of this vector species. Indeed, the authors find that Spiroplasma competes for finite nutrients negatively impacting female fecundity. Moreover in males Spiroplasma affects the motility of sperm impacting their transfer to female spermatheca.

These data validate the authors hypothesis and support the extreme importance and novelty of this work.

One point of weakness of this work is the way in which the authors describe the results (Results section). The data are presented in an overly discursive manner. This is a pity , as often this results in the authors and subsequently the reader losing the thread of the argument. The results should be presented in more efficient and succinct manner. Moreover, they include considerations that should be in the Discussion. This will avoid the repetitions that are present in Results and Discussion sections.

If this weakness is resolved, the paper is very important and should be considered for publication.

**Part II – Major Issues: Key Experiments Required for Acceptance**

Reviewer #1: My main concern is that authors draw their conclusion without accounting for other symbionts infecting the tsetse flies, which could have confounding effects. The obligate endosymbiont Wigglesworthia seems especially important in this case because of the nutritional complementation it provides to the fly (hence its effect on the fly reproductive fitness). I suggest authors to control for Wigglesworthia titer in their lab reared flies to determine whether the observed effects on fitness (i.e. increased length of the GC in females and decreased sperm motility in males) are due to a direct effect of Spiroplasma (if Wigg. titer remains unchanged upon Spiroplasma infection) or an indirect effect on the primary symbiont (if Wigg. titer decreases upon Spiroplasma infection).

Reviewer #2: I'd like to see qPCR performed to confirm the endpoint PCR underpinning the transmission experiments, and also the RNAseq (though here it may be better instead to check for Spiroplasma reads in the RNAseq reads).

I don't think it is feasible really for a revision, but would like to see a more controlled RNAseq using the lab colony to compare against the wild results - or some validation in lab colonies with qRT PCR. it would provide evidence of repeatable patterns and lack of confound.

Reviewer #3: The experimental planning is clear although it is too descriptive. The authors must justify why for some experiments they use a very reduced number of replicates such as, for instance, the case of the sperm motility test and other tests.

**Part III – Minor Issues: Editorial and Data Presentation Modifications**

Reviewer #1: Line 282 : accessary -> accessory.

Line 427 : the second occurence of GC2 on this line should be GC1.

Line 515 : the reference ([47]) seems misplaced.

Line 682 : my understanding is that pgrp-la has been used in this work as a housekeeping gene because has been used previously in the tsetse fly. In Drosophila however, this gene is immune-inducible, and the authors detected a mild Toll pathway activation the transcriptomics data upon Spiroplasma infection, which makes me wonder whether pgrp-la is a suitable housekeeping gene in this particular case. Can authors demonstrate that its expression is unchanged upon Spiroplasma infection ? If not, I recommend using an alternative gene for normalisation.

Fig 1B : I find this figure a bit difficult to grasp because sample details are provided only on the horizontal lines. Adding the symetrical annotation at the bottom of the columns could help readers to assess the clustering more easily.

Fig 3A : I find the figure display a bit confusing and would prefer to see the individual points instead of boxplots, as it has been made for the two other panels.

Same comment goes for Fig 4C, where it is hard to see the individual points on the violin plots.

Fig 3B : please indicate which correction has been applied to the multiple testing.

Supplementary data 1 and 2 : only tsetse-specific gene codes are indicated, which makes it very difficult for people from other fields to make use of these data. It would be helpful to add more indication on each gene, at least a column with their full name (or the name of their closest known homolog).

Reviewer #2: L97-99 This Spiroplasma does not have an official name, I think, so avoid raising one here? Microbiological nomenclature has rules for naming stuff. Also, it isn't really a close relative of S. poulsonii - makes it sound monophyletic with it - it is in the next clade along really. If one looks at a phylogeny, the first thing you say is not that it is a S. poulsonii relative.

L266 comprised not compromised.

L406 and elsewhere: can we avoid using moms and dads? Mothers and Fathers, or Male and female parents.

L416 and elsewhere: Table legends need improving - more detail to allow the table to be read standalone please.

L466-468: bit of a logical contrast - sex dimorphism and expression in reproductive tissues in sex biased gene expression are sort of opposites. Dimorphism is a body wide thing where the same tissues express differently, reproductive tissues are different tissues in the two sexes behaving differently. Sex biased gene expression between reproductive tissues is inevitable, but not necessarily linked to overall sexual dimorphism. Overall sex dimorphism and sex biased gene expression depends on the differences in male and female somatic phenotype - which varies between species.

L527: this logic is actually not true if there is paternal inheritance; if a symbiont has biparental inheritance it can invade the population without benefit (see Drosophila sigma virus as an example).

L537: did immunity fall out of the RNAseq analysis?

L548-551: this argument makes absolutely no logical sense to me, sorry. Higher VT maintains symbionts more efficiently than low VT.

L644 16S not 16s- S is a unit - the Svedberg - and is capitalised.

Reviewer #3: A major issue is the way in which the results are described. They must be much more concise in order that the data appear clear in their logical sequence and biological meaning.

PLOS authors have the option to publish the peer review history of their article (what does this mean?). If published, this will include your full peer review and any attached files.

Reviewer #1: No

Reviewer #2: No

Reviewer #3: No
---

## [Editor Report · Decision Letter 1]

7 Sep 2021

Dear Dr. Weiss,

We are pleased to inform you that your manuscript 'Infection with endosymbiotic Spiroplasma disrupts tsetse (Glossina fuscipes fuscipes) metabolic and reproductive homeostasis' has been provisionally accepted for publication in PLOS Pathogens.

Best regards,

Bruno Lemaitre

Guest Editor

PLOS Pathogens

Kirk Deitsch

Section Editor

PLOS Pathogens

Kasturi Haldar

Editor-in-Chief

PLOS Pathogens

orcid.org/0000-0001-5065-158X

Michael Malim

Editor-in-Chief

PLOS Pathogens

orcid.org/0000-0002-7699-2064

The paper has been improved and is now acceptable a publication
---

## [Editor Report · Acceptance letter]

10 Sep 2021

Dear Dr. Weiss,

We are delighted to inform you that your manuscript, "Infection with endosymbiotic </i>Spiroplasma</i> disrupts tsetse (</i>Glossina fuscipes fuscipes</i>) metabolic and reproductive homeostasis," has been formally accepted for publication in PLOS Pathogens.

Best regards,

Kasturi Haldar

Editor-in-Chief

PLOS Pathogens

orcid.org/0000-0001-5065-158X

Michael Malim

Editor-in-Chief

PLOS Pathogens

orcid.org/0000-0002-7699-2064